# A Fast Intersection of Confidence Intervals Method-Based Adaptive Thresholding for Sparse Image Reconstruction Using the Matrix Form of the Wavelet Transform

Ivan Volaric *[ID] and Victor Sucic [ID]

Faculty of Engineering, University of Rijeka, HR-51000 Rijeka, Croatia; viktor.sucic@riteh.uniri.hr
* Correspondence: ivan.volaric@riteh.uniri.hr

**Abstract:** One of the frequently used classes of sparse reconstruction algorithms is based on the iterative shrinkage/thresholding procedure, in which the thresholding parameter controls a trade-off between the algorithm's accuracy and execution time. In order to avoid this trade-off, we propose using a fast intersection of confidence intervals method in order to adaptively control the threshold value throughout the iterations of the reconstruction algorithm. We have upgraded the two-step iterative shrinkage thresholding algorithm with a such procedure, improving its performance. The proposed algorithm, denoted as the FICI-TwIST, along with a few selected state-of-the-art sparse reconstruction algorithms, has been tested on the classical problem of image recovery by emphasizing the image sparsity in the discrete cosine and the discrete wavelet domain. Furthermore, we have derived a single wavelet transformation matrix which avoids wrapping effects, thereby achieving significantly faster execution times as compared to a more traditional function-based transformation. The obtained results indicate the competitive performance of the proposed algorithm, even in cases where all algorithm parameters have been individually fine-tuned for best performance.

**Keywords:** compressive sensing; fast intersection of confidence intervals; image reconstruction; iterative soft thresholding; signal sparsity; sparse reconstruction algorithm



## 1. Introduction

Signal sparsity is a signal property that has been used for many decades, mostly for lossy multimedia compression [1]. A signal is considered *K*-sparse when most of its energy is located in only *K* samples and most of the signal samples are close to zero. In most cases, signals do not exhibit sparsity in the observation domain (e.g., temporal or spatial); however, they are sparse in some alternative domain, which enables lossy compression by discarding the low-energy samples. The discrete cosine transform (DCT) and the discrete wavelet transform (DWT) are examples of sparsity-inducing transformations that are widely used in popular file formats, such as JPEG, JPEG2000, MP3, etc.

Compressive sensing (CS) is a newly developing paradigm that has gained the interest of many researchers in the last decade, especially after ground-breaking papers [2,3]. It also exploits the signal sparsity but in a slightly different way. Instead of recording a full data-set, only a small subset of the original data is recorded, while the missing samples are calculated in a way that will produce the sparsest representation in the a priori known sparse domain. In real-life situations, the partial unavailability of samples can happen due to various physical constraints or corrupted data, resulting in wide-spread application of the CS-based methods in multimedia [1,4–7], medicine [8,9], geoscience [10,11], radar [12,13], wireless communication [14,15], etc.

In this paper, we continue our previous research [6,7], in which we have augmented the two-step iterative shrinkage thresholding (TwIST) algorithm [4] with the fast intersection of confidence intervals (FICI) method [16], providing a data-driven threshold value for the TwIST algorithm. This modification has resulted in the sparse reconstruction algorithm,

denoted as the FICI-TwIST, in which we have removed the dependency on the user-defined threshold value that controls a trade-off between the solution accuracy and the convergence rate of a considered class of the reconstruction algorithms. In this paper, we have derived a single-wrapped Cohen–Daubechies–Feauveau 9/7 (CDF97) matrix, which enabled us to test the previously proposed algorithm not only by emphasizing the sparsity in the DCT domain but also in the DWT domain. Using such a matrix, which to the best of our knowledge, cannot be found in the literature, significantly decreased the algorithm execution times, as shown in this paper.

The rest of the paper is organized as follows. Section 2 gives the theoretical background behind sparse image reconstruction, derives the used transformation matrices, and introduces the proposed reconstruction algorithm, while Section 3 presents the simulation results. Section 4 gives the concluding remarks.

## 2. Sparse Image Reconstruction

This section introduces the theory behind the proposed method and is organized as follows. Section 2.1 introduces the problem of sparse image reconstruction, while, in Section 2.2, we discuss a method for solving it that uses $\ell_1$-norm minimization. Section 2.3 presents the DCT and the DWT transformation matrices used to solve this problem, while Section 2.4 briefly describes our previous work [6,7] regarding the FICI-TwIST algorithm.

### 2.1. Problem Formulation

Let $x$ denote a grey-scale image with $N_x \times N_y$ pixels, and let $\boldsymbol{\Phi}$ denote an invertible linear transformation, such that:

$$X = \boldsymbol{\Phi}^{(N_x)} \cdot x \cdot \boldsymbol{\Phi}^{(N_y)^T}, \tag{1}$$

which results in $X$ being $K$-sparse, where $K \ll N_x N_y$. For more convenient mathematical notation, we will rewrite (1), where both $x_V = \text{vect}(x)$ and $X_V = \text{vect}(X)$ are column vectors, as:

$$X_V = \boldsymbol{\Phi}_{1D} \cdot x_V, \tag{2}$$

where $\boldsymbol{\Phi}_{1D}$ is the $N_x N_y \times N_x N_y$ transformation matrix performing an equivalent transformation to that in (1). Section 2.3 provides a more detailed discussion about the used transformation matrices $\boldsymbol{\Phi}$, their 1D equivalents $\boldsymbol{\Phi}_{1D}$, and their inverses.

Let $y_V = \boldsymbol{\Psi} x_V$ denote a column vector of $M \ll N_x N_y$ available image pixels randomly selected from $x_V$, where $\boldsymbol{\Psi}$ is the $M \times N_x N_y$ measurement matrix, and each row contains a single entry with a value of one (the rest being zeroes), connecting a $y_V$ $m^{\text{th}}$ available pixel with a $x_V$ $n^{\text{th}}$ image pixel. By combining $y_V$ with (2), we define the sparse reconstruction problem:

$$y_V = \boldsymbol{\Psi} x_V = \boldsymbol{\Psi} \boldsymbol{\Phi}_{1D}^{-1} X_V = \tilde{A} X_V, \tag{3}$$

where $\tilde{A} = \boldsymbol{\Psi} \boldsymbol{\Phi}_{1D}^{-1}$ is the truncated backward transformation matrix with deleted rows corresponding to the missing samples. In a similar way, matrix $A = \boldsymbol{\Phi}_{1D} \boldsymbol{\Psi}^T$, used in future expressions, represents the expanded forward transformation, where the missing samples are set to zero prior to the forward transformation. Since matrix $\tilde{A}$ is not invertible, the following unconstrained optimization problem has to be solved [4,5]:

$$\widehat{X}_V = \arg \min_{X_V} \left\{ \frac{1}{2} ||X_V - A y_V||_2^2 + \lambda c(X_V) \right\}, \tag{4}$$

where $c(X_V) : \mathbb{R}^{N_x \times N_y} \to \mathbb{R}$ is the regularization function, while $\lambda$ is the regularization parameter. The first term can be interpreted as the error-measuring function weighted by one-half, while the second term, weighted by $\lambda$, measures the signal property that we want to attain through the optimization procedure. Because of this, $\lambda$ can be interpreted as a parameter that controls the solution accuracy: for larger $\lambda$ values, the second term becomes

more important in the minimization, that is, the first term (i.e., the reconstruction error) becomes less important [4,5].

### 2.2. Problem Solution with the $\ell_1$-Norm Minimization

The regularization function plays the most significant role in the previously described optimization problems, as it is a function to be minimized. As already stated, its role is to emphasize the a priori known solution property, which, in this case, is the signal sparsity. In other words, minimizing $c(X_V)$ should maximize the number of zero-valued samples in $X_V$. The best function for this task is the $\ell_0$-norm, as it counts the number of non-zero samples. However, the downside of the $\ell_0$-norm minimization is that it is an NP-hard combinatorial problem, usually solved with greedy algorithms, by searching for a good local minima instead of the global one [17]. Because of this, this problem is often relaxed with the easier to solve convex problem of the $\ell_1$-norm minimization [4,5,17,18], thereby introducing a new problem: the objective function does not measure the exact signal property that we want to attain. A more detailed survey of the sparse reconstruction algorithms is given in [1,19,20].

By using the $\ell_1$-norm-based regularization function, we can rewrite (4) as:

$$X_V^{\ell_1} = \arg\min_{X_V} ||X_V||_1, \text{ s. t.: } ||X_V - Ay_V||_2^2 \le \epsilon, \tag{5}$$

allowing a small difference, $\epsilon$, between the available pixels and their reconstructed counterparts in order to account for noise. This expression can be further simplified by using the Moreau proximity operator [21]:

$$X_V^{\ell_1} = \text{soft}_\lambda\{X_V\} = \text{sgn}(X_V)\max\{|X_V| - \lambda, 0\}. \tag{6}$$

Note that the soft-thresholding parameter $\lambda$ is the regularization parameter from (4), and, in the context of (6), it can be interpreted in a similar fashion: a parameter that controls a trade-off between the solution accuracy and the convergence rate. With a higher $\lambda$ value, the input signal is going to be thresholded more strictly, resulting in a lower accuracy and a faster convergence rate, and vice versa [4,5].

An example of the sparse reconstruction algorithm that achieves the $\ell_1$-norm minimization through iterative soft-thresholding is the TwIST algorithm [4]:

$$\left[X_V^{\ell_1}\right]^{[n+1]} = (1-\alpha)\left[X_V^{\ell_1}\right]^{[n-1]} + (\alpha - \beta)\left[X_V^{\ell_1}\right]^{[n]} + \beta\,\text{soft}_\lambda\left\{\left[X_V^{\ell_1}\right]^{[n]} + A\left(y_V - \tilde{A}\left[X_V^{\ell_1}\right]^{[n]}\right)\right\}, \tag{7}$$

where $\alpha$ and $\beta$ are the user-defined TwIST relaxation parameters controlling the averaging weights between the current and the previous two solutions. The final solution is obtained by iterating (7) until the stopping criterion is satisfied. In this paper, we have used two stopping criteria: (1) the relative change in the $\ell_2$-norm between the solution of two consecutive algorithm iterations drops below $\epsilon_{\ell_2}$, or (2) the maximum number of iterations, $N_{it}$, has been reached.

### 2.3. DCT and DWT Transformation Matrices

The DCT matrix is given by:

$$\mathbf{\Phi}_{k,n}^{(N)} = \begin{cases} \sqrt{\frac{1}{N}}, & k = 0, \\ \sqrt{\frac{2}{N}}\cos\left(\frac{\pi(2n+1)k}{2N}\right), & \text{otherwise,} \end{cases} \tag{8}$$

for $k, n \in [0, \ldots, N-1]$. The transformation matrix, $\boldsymbol{\Phi}_{1D}$, used in (2), is then simply defined through the Kronecker product as:

$$
\boldsymbol{\Phi}_{1D} = \boldsymbol{\Phi}^{(N_x)} \otimes \boldsymbol{\Phi}^{(N_y)} = \begin{bmatrix} \boldsymbol{\Phi}_{1,1}^{(N_x)}\boldsymbol{\Phi}^{(N_y)} & \boldsymbol{\Phi}_{1,2}^{(N_x)}\boldsymbol{\Phi}^{(N_y)} & \cdots & \boldsymbol{\Phi}_{1,N_x}^{(N_x)}\boldsymbol{\Phi}^{(N_y)} \\ \boldsymbol{\Phi}_{2,1}^{(N_x)}\boldsymbol{\Phi}^{(N_y)} & \boldsymbol{\Phi}_{2,2}^{(N_x)}\boldsymbol{\Phi}^{(N_y)} & \cdots & \boldsymbol{\Phi}_{2,N_x}^{(N_x)}\boldsymbol{\Phi}^{(N_y)} \\ \vdots & \vdots & \ddots & \vdots \\ \boldsymbol{\Phi}_{N_x,1}^{(N_x)}\boldsymbol{\Phi}^{(N_y)} & \boldsymbol{\Phi}_{N_x,2}^{(N_x)}\boldsymbol{\Phi}^{(N_y)} & \cdots & \boldsymbol{\Phi}_{N_x,N_x}^{(N_x)}\boldsymbol{\Phi}^{(N_y)} \end{bmatrix}. \tag{9}
$$

Note that $\boldsymbol{\Phi}$ is real and orthonormal for the DCT; thus, inverse calculation is not needed since $\boldsymbol{\Phi}^{-1} = \boldsymbol{\Phi}^{T}$. $\boldsymbol{\Phi}_{1D}$ follows the same property; thus, $\boldsymbol{\Phi}_{1D}^{-1} = \boldsymbol{\Phi}_{1D}^{T}$, and, most importantly, $\tilde{\boldsymbol{A}} = \boldsymbol{A}^{T}$.

Unlike the DCT, the DWT matrix is more complex to construct; hence, it is usually represented with the filter banks. The analysis bank performs forward transformation, while the synthesis bank performs backward transformation. In the analysis bank, the approximation vector at the 1st scale is created by filtering the input vector with a low-pass filter, while the detail vector is obtained by filtering with a high-pass filter, followed by down-sampling with a factor of two. In this fashion, all subsequent scales perform filtering and down-sampling of the approximation vector from the previous scale, ultimately resulting in the *l*-th scale approximation vector and the 1st–*l*-th scale detail vectors. In the synthesis bank, the approximation and detail vector of the *l*-th scale are up-sampled by a factor of two, respectively filtered with a low-pass and a high-pass filter, and summed, creating the approximation vector of the $(l-1)$-th scale. When the input signal is 2D, both filters are applied both row- and column-wise, decomposing it into the approximation and three detail matrices (vertical, horizontal, and diagonal) with every subsequential scale further decomposing only the approximation matrix.

In order to construct the CDF97 matrix of the *l*-th scale, let us start with the pair of biorthogonal low-pass filters of lengths $L_{\tilde{h}} = 9$ and $L_h = 7$, with the following coefficients, already pre-scaled by a factor of $\sqrt{2}$ and $1/\sqrt{2}$, respectively [22]:

$$
\begin{array}{rclrrcl}
\tilde{h}(0) &=& 0.852699, & h(0) &=& 0.788486, \\
\tilde{h}(-1) = \tilde{h}(1) &=& 0.377403, & h(-1) = h(1) &=& 0.418092, \\
\tilde{h}(-2) = \tilde{h}(2) &=& -0.110624, & h(-2) = h(2) &=& -0.040689, \\
\tilde{h}(-3) = \tilde{h}(3) &=& 0.023850, & h(-3) = h(3) &=& -0.064539, \\
\tilde{h}(-4) = \tilde{h}(4) &=& 0.037829. & & &
\end{array} \tag{10}
$$

By using the $\tilde{h}(k)$ filter coefficients, we can construct a low-pass portion of the transformation matrix on the *l*-th scale:

$$
\tilde{H}_{k,n}^{(N,l)} = \begin{cases} \begin{cases} \tilde{h}(m_1) + \tilde{h}(m_2), & c_2 = \text{true}, \\ \tilde{h}(m_1) + \tilde{h}(m_3), & c_3 = \text{true}, \\ \tilde{h}(m_1), & c_2 = c_3 = \text{false}, \end{cases} & c_1 = \text{true}, \\ 0, & c_1 = \text{false}, \end{cases} \tag{11}
$$

for $k \in \left[0, \ldots, \frac{N}{2^l} - 1\right]$, $n \in \left[0, \ldots, \frac{N}{2^{l-1}} - 1\right]$, and where $m_1 = n - 2k$, $m_2 = -n - 2k$, $m_3 = \frac{N}{2^{l-2}} + m_2 - 2$, while the conditions $c_1$–$c_3$ are listed in the first row of Table 1. Such definition of the wavelet matrix negates the wrapping effects, caused by the filter coefficients wrapping around the matrix edges, avoiding the need for input signal periodization. The first condition sums the coefficients in the last columns of the first $\lceil (L_{\tilde{h}} - 1)/4 \rceil$ rows with the appropriate coefficients in the first columns. In the similar way, the second condition sums the coefficients in the first columns of the last $\lfloor L_{\tilde{h}}/4 \rfloor$ rows with the appropriate coefficients in the last columns. These two conditions, involving indices $m_2$ and $m_3$, are only valid for a handful of elements, mainly in the top left and bottom right corners, while most of the coefficients are calculated just as $\tilde{h}(m_1)$.

**Table 1.** Index restrictions for DWT matrices.

| | $c_1$ | $c_2$ | $c_3$ |
|---|---|---|---|
| $\tilde{H}^{(N,l)}$ | $m_1 \leq \frac{L_{\tilde{h}}-1}{2}$ | $m_2 \leq \frac{L_{\tilde{h}}-1}{2}$ && $n \neq 0$ | $m_3 \leq \frac{L_{\tilde{h}}-1}{2}$ && $n \neq \frac{N}{2^{l-1}} - 1$ |
| $\tilde{G}^{(N,l)}$ | $m_1 - 1 \leq \frac{L_h-1}{2}$ | $m_2 - 1 \leq \frac{L_h-1}{2}$ && $n \neq 0$ | $m_3 - 1 \leq \frac{L_h-1}{2}$ && $n \neq \frac{N}{2^{l-1}} - 1$ |
| $H^{(N,l)}$ | $m_1 \leq \frac{L_h-1}{2}$ | $m_2 \leq \frac{L_h-1}{2}$ && $k \neq 0$ | $m_3 \leq \frac{L_h-1}{2}$ |
| $G^{(N,l)}$ | $m_1 - 1 \leq \frac{L_{\tilde{h}}-1}{2}$ | $m_2 - 1 \leq \frac{L_{\tilde{h}}-1}{2}$ | $m_3 - 1 \leq \frac{L_{\tilde{h}}-1}{2}$ && $k \neq \frac{N}{2^l} - 1$ |

Note, however, such matrix definition restricts the DWT scale in two ways: (1) $N_x$ and $N_y$ have to be divisible by $2^l$, and (2) $\min\{N_x, N_y\}/2^{l-1} \geq \max\{L_{\tilde{h}}, L_h\}$. The first condition is easily avoided by zero-padding; however, the second one seriously limits the DWT scale. Since the DWT domain becomes sparses as the DWT scale increases, it is imperative to avoid this restriction. To better understand (11), the coefficient placement in the $k$-th matrix row is depicted by Figure 1. The second limiting factor ensures that the depicted coefficient wrapping happens no more than once per row, that is, that no more than two coefficients are summed, since (11) requires such a condition. However, using the same logic of turning back when the first or last column is reached, with multiple turns, we can place all of the filter coefficients regardless the number of columns, resulting in the matrix entries, which are sums of more than two filter coefficients.

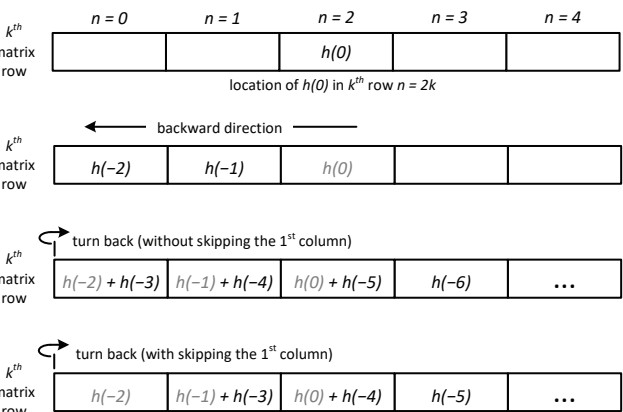

**Figure 1.** Placement of the filter coefficients in the $k$-th matrix row. The first column is skipped based on the conditions listed in Table 1. Only the placement of negative indices is depicted, however, the same logic applies for positive indices, which are summed with the existing entries.

In a similar fashion, we can design a high-pass portion of the transformation matrix on the $l$-th scale. Filter coefficients are calculated as $\tilde{g}(k) = (-1)^k h(1-k)$, and matrix $\tilde{G}^{(N,l)}$ is generated analogous to (11), with the conditions $c_1$–$c_3$ listed in the second row of Table 1. The only difference is caused by the indexing difference between $\tilde{h}_k$ and $\tilde{g}_k$, since the $\tilde{g}_k$ coefficients are shifted by one.

Matrix $\left[ \tilde{H}^{(N,l)T} \mid \tilde{G}^{(N,l)T} \right]^T$ performs a single scale transformation from the $(l-1)$-th scale to the $l$-th scale. In order to create a complete transformation matrix from the 0th to the $l$-th scale we need to consecutively multiply corresponding matrices, defining the complete matrix representation of the analysis filter bank [23]:

$$\mathbf{\Phi}^{(N)} = \begin{bmatrix} \tilde{H}^{(N,l)} \tilde{H}^{(N,l-1)} \cdots \tilde{H}^{(N,1)} \\ \tilde{G}^{(N,l)} \tilde{H}^{(N,l-1)} \cdots \tilde{H}^{(N,1)} \\ \tilde{G}^{(N,l-1)} \tilde{H}^{(N,l-2)} \cdots \tilde{H}^{(N,1)} \\ \vdots \\ \tilde{G}^{(N,2)} \tilde{H}^{(N,1)} \\ \tilde{G}^{(N,1)} \end{bmatrix}. \tag{12}$$

However, due to the image's 2D nature, if we apply such matrix to (1), or use the Kronecker product (9) for (2), the result will differ from the traditional DWT definition. Not only is the approximation sub-matrix going to be decomposed in all subsequent DWT scales, but the horizontal and vertical detail sub-matrices will be as well. Although not 'wrong', we wanted to maintain the traditional DWT definition, thus $\mathbf{\Phi}_{1D}$ is calculated by applying the Kronecker product for each sub-matrix block separately [23]:

$$
\mathbf{\Phi}_{1D} = \begin{bmatrix}
\left( \tilde{\boldsymbol{H}}^{(N_y,l)} \tilde{\boldsymbol{H}}^{(N_y,l-1)} \dots \tilde{\boldsymbol{H}}^{(N_y,1)} \right) \otimes \left( \tilde{\boldsymbol{H}}^{(N_x,l)} \tilde{\boldsymbol{H}}^{(N_x,l-1)} \dots \tilde{\boldsymbol{H}}^{(N_x,1)} \right) \\
\left( \tilde{\boldsymbol{H}}^{(N_y,l)} \tilde{\boldsymbol{H}}^{(N_y,l-1)} \dots \tilde{\boldsymbol{H}}^{(N_y,1)} \right) \otimes \left( \tilde{\boldsymbol{G}}^{(N_x,l)} \tilde{\boldsymbol{H}}^{(N_x,l-1)} \dots \tilde{\boldsymbol{H}}^{(N_x,1)} \right) \\
\left( \tilde{\boldsymbol{G}}^{(N_y,l)} \tilde{\boldsymbol{H}}^{(N_y,l-1)} \dots \tilde{\boldsymbol{H}}^{(N_y,1)} \right) \otimes \left( \tilde{\boldsymbol{H}}^{(N_x,l)} \tilde{\boldsymbol{H}}^{(N_x,l-1)} \dots \tilde{\boldsymbol{H}}^{(N_x,1)} \right) \\
\left( \tilde{\boldsymbol{G}}^{(N_y,l)} \tilde{\boldsymbol{H}}^{(N_y,l-1)} \dots \tilde{\boldsymbol{H}}^{(N_y,1)} \right) \otimes \left( \tilde{\boldsymbol{G}}^{(N_x,l)} \tilde{\boldsymbol{H}}^{(N_x,l-1)} \dots \tilde{\boldsymbol{H}}^{(N_x,1)} \right) \\
\left( \tilde{\boldsymbol{H}}^{(N_y,l-1)} \tilde{\boldsymbol{H}}^{(N_y,l-2)} \dots \tilde{\boldsymbol{H}}^{(N_y,1)} \right) \otimes \left( \tilde{\boldsymbol{G}}^{(N_x,l-1)} \tilde{\boldsymbol{H}}^{(N_x,l-2)} \dots \tilde{\boldsymbol{H}}^{(N_x,1)} \right) \\
\left( \tilde{\boldsymbol{G}}^{(N_y,l-1)} \tilde{\boldsymbol{H}}^{(N_y,l-2)} \dots \tilde{\boldsymbol{H}}^{(N_y,1)} \right) \otimes \left( \tilde{\boldsymbol{H}}^{(N_x,l-1)} \tilde{\boldsymbol{H}}^{(N_x,l-2)} \dots \tilde{\boldsymbol{H}}^{(N_x,1)} \right) \\
\vdots \\
\tilde{\boldsymbol{G}}^{(N_y,1)} \otimes \tilde{\boldsymbol{H}}^{(N_x,1)} \\
\tilde{\boldsymbol{G}}^{(N_y,1)} \otimes \tilde{\boldsymbol{G}}^{(N_x,1)}
\end{bmatrix}. \tag{13}
$$

As in the DCT case, we again want to avoid the inverse calculation of the relatively large matrix $\mathbf{\Phi}_{1D}$. However, the DWT matrix is not orthonormal, thus $\mathbf{\Phi}^{-1} \neq \mathbf{\Phi}^T$, $\mathbf{\Phi}_{1D}^{-1} \neq \mathbf{\Phi}_{1D}^T$, and, most importantly, $\tilde{\boldsymbol{A}} \neq \boldsymbol{A}^T$. This is where the biorthogonal property helps. Although quite possible, we will not calculate the inverse in a traditional way, but rather we will construct a matrix $\mathbf{\Phi}_{1D}^{-1}$, using the same coefficients (10), with a similar procedure. The low-pass filter matrix, $\boldsymbol{H}^{(N,l)}$, is constructed from the coefficients $h(k)$, while the high-pass filter matrix, $\boldsymbol{G}^{(N,l)}$, is constructed from the coefficients $g(k) = (-1)^k \tilde{h}(1-k)$. Both matrices are generated analogous to (11), while the conditions $c_1$ - $c_3$ are listed in the last two rows of Table 1. While most of the coefficients are calculated in the same way (as $h(m_1)$), the wrapping effects in the synthesis filter bank are dealt column-wise, resulting in slight condition differences. The matrix $\mathbf{\Phi}_{1D}^{-1}$, representing the entire synthesis filter bank, is then constructed analogous to (13).

It is also worth mentioning that (11) is only valid for all four matrices if the filter coefficients are symmetric; in the backward mode, we did not take into account mirroring in the wrapping effects. Such compromise has been taken in order to provide the elegancy of a single expression that is valid for all four matrices. In more general terms, both $m_2$ and $m_3$ should be multiplied with $(-1)$ in the backward mode, while, in $\boldsymbol{G}^{(N,l)}$, both indices have an additional $(+2)$ shift after the multiplication.

### 2.4. FICI-Based Adaptive Thresholding

As already stated, the performance of the sparse reconstruction algorithms that achieves $\ell_1$ minimization through soft-thresholding is highly dependent on the regularization parameter $\lambda$ in (4), that is, the threshold value in (6), controlling a trade-off between the solution accuracy and the convergence rate. In order to achieve both benefits, $\lambda$ can be variable through the algorithm iterations: starting relatively high and decreasing as the algorithm converges towards the optimal solution. In our previous research [6,7], we proposed the FICI-TwIST algorithm, providing an adaptive threshold value calculation in every TwIST iteration. The FICI method searches the vicinity of the specific signal sample for a region with statistically similar amplitude values, which we have used in order to find a region with the statistically lowest amplitudes to be thresholded. The complete FICI-TwIST pseudo-code is given in Algorithm 1, while a more detailed discussion can be found in [6,7].

---

**Algorithm 1** The FICI-TwIST algorithm.

---

**Input:** $y_V, \Psi, \alpha, \beta, \Gamma, R_c, N_{reg}, \lambda_F, \epsilon_{\ell_2}, N_{it}$
**Output:** $x_V$

    calculate $A$ and $\tilde{A}$ as described in Section 2.3;

    $\left[X_V^{\ell_1}\right]^{[-1]}, \left[X_V^{\ell_1}\right]^{[0]} \leftarrow Ay_V$

    **for** $n_{it} = 0$ **to** $N_{it}$ **do**

        $\ddot{X}_V \leftarrow \text{sort}\left\{ \left[X_V^{\ell_1}\right]^{[n]} + A\left( y_V - \tilde{A}\left[X_V^{\ell_1}\right]^{[n]} \right) \right\}$

        $\dot{X}_V \leftarrow \text{soft}_{\lambda_F \max\{\ddot{X}_V\}}\{\ddot{X}_V\}$

        $i_0 \leftarrow$ index of the first non-zero entry in $\dot{X}_V$;

        **for** $n_{reg} = 1$ **to** $N_{reg}$ **do**

            $\Delta i \leftarrow 1$

            $R \leftarrow -1$

            **while** $R < R_c$ **do**

                mean $\leftarrow$ update the mean value of samples $\dot{X}_V(i_0), \ldots, \dot{X}_V(i_0 + \Delta i)$;

                std $\leftarrow$ recalculate the standard deviation of samples $\dot{X}_V(i_0), \ldots, \dot{X}_V(i_0 + \Delta i)$;

                $D_{u,l} \leftarrow$ mean $\pm \Gamma$ std;

                $D_{u_{\min}} \leftarrow \min\{D_u, D_{u_{\min}}\}$;

                $D_{l_{\max}} \leftarrow \max\{D_l, D_{l_{\max}}\}$;

                $R \leftarrow \frac{D_{u_{\min}} - D_{l_{\max}}}{2\Gamma \text{ std}}$;

                $\Delta i \leftarrow \Delta i + 1$

            **end while**

            $i_0 \leftarrow i_0 + \Delta i$

        **end for**

        $\lambda \leftarrow \dot{X}_V(i_0)$

        $\left[X_V^{\ell_1}\right]^{[n_{it}+1]} \leftarrow$ (7)

        **if** $\left| \frac{\left|\left|\left[X_V^{\ell_1}\right]^{[n_{it}]}\right|\right|_2^2 - \left|\left|\left[X_V^{\ell_1}\right]^{[n_{it}+1]}\right|\right|_2^2}{\left|\left|\left[X_V^{\ell_1}\right]^{[n_{it}+1]}\right|\right|_2^2} \right| \leq \epsilon_{\ell_2}$ **then break**

    **end for**

    **return** $x_V = \Phi_{1D}^{-1}\left[X_V^{\ell_1}\right]^{[n_{it}+1]}$

---

## 3. Results and Discussion

    The reconstruction performance of the proposed algorithm was tested on a standard grey-scale test image Lenna, pre-scaled to $256 \times 256$ pixels for both the DCT and the DWT as the sparsity-inducing transformations in two scenarios: (1) the image was divided into $8 \times 8$ blocks with each block processed individually and (2) without the block division. The DWT scale was set to its maximum: (1) $L = 3$ and (2) $L = 8$ in order to produce the sparsest domain. In the first scenario, the transformations were implemented exactly as described in Section 2.3, while the second scenario implemented the DWT with a lifting scheme, and the DCT using the MATLAB build-in function, since the $256^2 \times 256^2$ transformation matrix would require 16GB of memory space using a single precision float.

    The reconstruction performance was evaluated in terms of the mean square error (MSE) and the algorithm's execution time, while the FICI-TwIST algorithm was compared with the following state-of-the-art reconstruction algorithms: the TwIST [4], the Split-augmented Lagrangian shrinkage algorithm (SALSA) [5], the Nesterov algorithm (NESTA) [18], and the your-augmented Lagrangian algorithm for $\ell_1$ (YALL1) [17]. The simulations were performed for a range of CS ratios, $M/N_x N_y \in [0.1, \ldots, 0.9]$, while the stopping criteria were set to $\epsilon_{\ell_2} = 10^{-5}$ and $N_{it} = 1000$. For each CS ratio, the results were averaged over $N_{CS} = 50$ runs with the randomly generated measuring matrices. All algorithm parameters were fine-tuned for best performance with the CS ratio of 0.4, which, in hindsight, did

not highlight the main advantage of the proposed algorithm: its adaptivity; however, 'sabotaging' other algorithms would be, lightly said, dishonest.

The obtained MSE values are presented in Figure 2, while the algorithm execution times are presented in Figure 3. When the image is divided into blocks, SALSA (for the DCT) and TwIST (for the DWT) run significantly worse than the other algorithms for lower CS ratios. However, in general, all algorithms have very similar reconstruction performances, likely due to the mentioned parameter fine-tuning. In the block scenarios, the FICI-TwIST runs very similarly to the NESTA, almost completely overlapping over the entire CS range. These algorithms emerge as the most successful algorithms. In the DCT case, the FICI-TwIST is slightly inferior, while, in the DWT case, it is slightly better. When the image is not divided into blocks, the FICI-TwIST, in general, is the second best for lower CS ratios and the worst for higher CS ratios. In both scenarios, our simulations have shown that the DCT outperforms the DWT in MSE terms. This is especially noticeable in the block scenario, for which a possible explanation is that the selected block size results in a less sparse transformation due to the DWT scale limit.

The algorithm execution times are, in general, constant over the entire CS range, having relatively similar ratios between the algorithms in all four scenarios. The FICI-TwIST runs the slowest, while the TwIST algorithm runs the fastest. This was to be expected, since FICI-TwIST requires constant re-calculation of the mean value and the standard deviation over the increasing window length, while TwIST is a relatively simple algorithm. This fact can be alleviated by skipping the threshold calculation in some of the FICI-TwIST iterations. In the block scenarios, there is very little difference in the algorithm execution times between the DCT and DWT, which was expected, since both transformations involve matrix multiplication. In addition, the block scenario is, in general, two times faster, regardless of the algorithm and the sparsity-inducing domain. Due to our simulation setup, it is hard to distinguish between the impacts of different implementations vs. the block size on this fact; however, using function-based transformation in the block scenario increased the execution times by a factor of $\{5, 25, 50, 15, 4\}$ and $\{20, 20, 15, 12, 12\}$, respectively, for the algorithm and domain. It is also noteworthy that the FICI-TwIST exhibits a strange behavior: in the DCT case, there is very little difference in the execution times between the block and non-block scenario, resulting in the second-best execution time for the non-block scenario. On the other hand, in the non-block DWT scenario, the execution time started to increase in the middle CS ratios.

In our simulations, the CS ratio of 0.4 was shown to be borderline, with the lower ratio resulting in a visibly significantly worse reconstruction performance, while, on the other hand, a higher CS ratio resulted in visually indistinguishable images. This fact guided us to fine-tune parameters for this specific ratio, and this is why we have shown the algorithm convergence rate (in terms of the MSE) and cumulative execution times over algorithm iterations in Figures 4 and 5, respectively. Figure 5 reveals that all iterations within a specific algorithm take relatively similar amounts of time, with the exception of the proposed algorithm in the DWT block scenario. In all cases, the MSE value settles between the 150th and 200th iteration; however, only TwIST and NESTA did not exit due to the iteration limit, with all other algorithms running full $N_{it} = 1000$ iterations. This reveals a shortcoming of the selected exit criterion (the relative $\ell_2$-norm change) and a possibility for decreasing the execution times by selecting some other, more appropriate criteria. However, in real-life implementation, not having access to the original image limits the design space for such a criterion. Moreover, the proposed algorithm in the block scenarios experiences the second-best convergence rate in the starting iterations, with only YALL1 having a better MSE improvement per iteration.

A single, randomly selected run with the CS ratio of 0.4 is shown in Figures 6–9 for all four cases. By visual inspection, we can confirm that all scenarios run very competitively, with the blocked DWT being an outlier due to the aforementioned DWT scale limit. In this scenario, the TwIST algorithm did not converge for some of the blocks. If we compare Figures 6 and 9, in our opinion, of the two best visually performing cases, Figure 6 (blocked

DCT) seems to be a little more sharper, revealing more of the reconstruction defects, while Figure 9 (whole DWT) is much smoother and (perhaps) visually better looking. However, we leave the task of subjective image quality assessment to the reader, since such task is hard, if not impossible, to numerically evaluate.

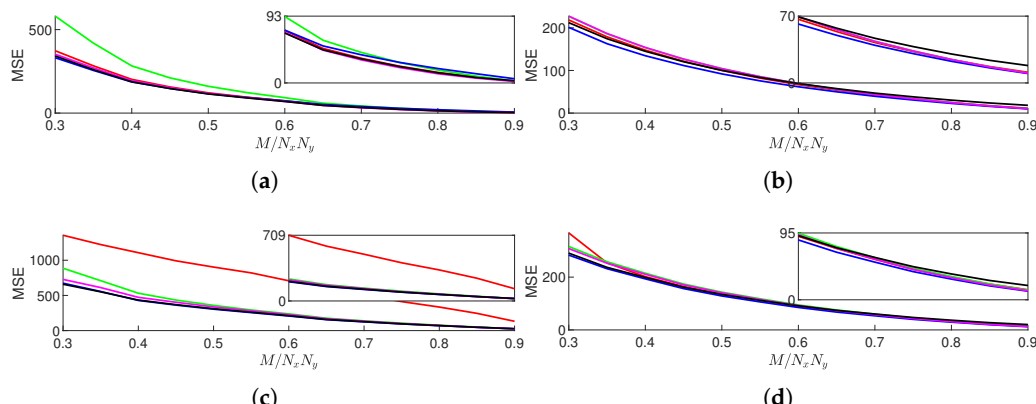

**Figure 2.** Average reconstructed MSE values for TwIST (red), SALSA (green), NESTA (blue), YALL1 (magenta), and FICI-TwIST (black) over $M/N_x N_y$ range; and scenario: (**a**) DCT blocked, (**b**) DCT whole, (**c**) DWT blocked, and (**d**) DWT whole. Range of the zoom inset is $M/N_x N_y \in [0.6, 0.9]$.

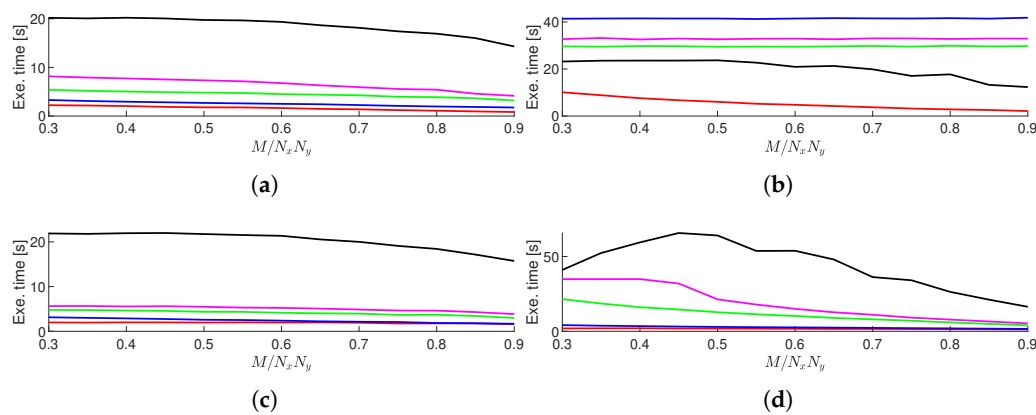

**Figure 3.** Average algorithm execution time for TwIST (red), SALSA (green), NESTA (blue), YALL1 (magenta), and FICI-TwIST (black) over $M/N_x N_y$ range; and scenario: (**a**) DCT blocked, (**b**) DCT whole, (**c**) DWT blocked, and (**d**) DWT whole.

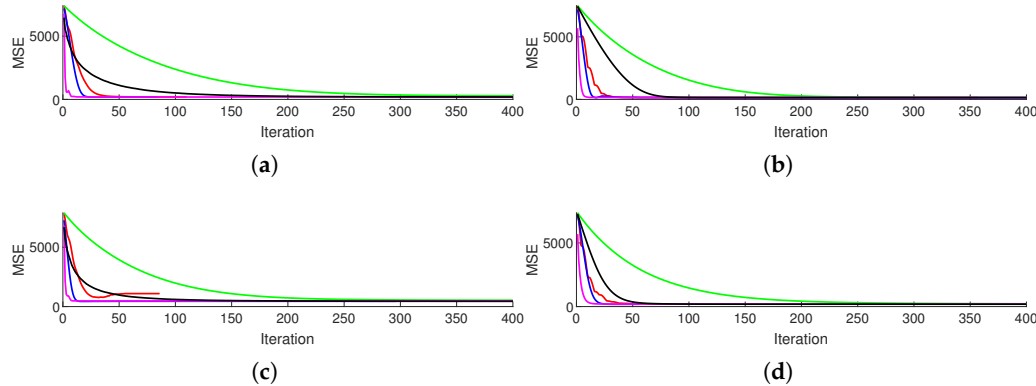

**Figure 4.** Average reconstructed MSE values ($M/N_x N_y = 0.4$) for TwIST (red), SALSA (green), NESTA (blue), YALL1 (magenta), and FICI-TwIST (black) over algorithm iterations; and scenario: (**a**) DCT blocked, (**b**) DCT whole, (**c**) DWT blocked, and (**d**) DWT whole.

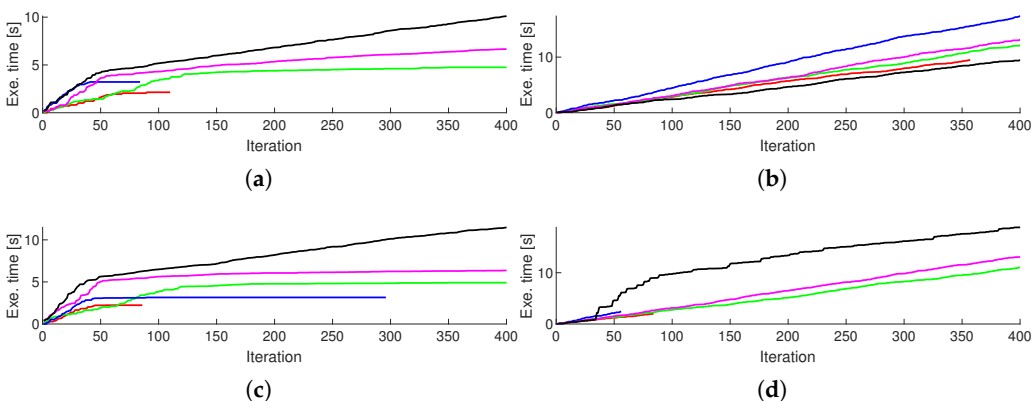

(a)

(b)

(c)

(d)

**Figure 5.** Average cumulative algorithm execution times ($M/N_x N_y = 0.4$) for TwIST (red), SALSA (green), NESTA (blue), YALL1 (magenta), and FICI-TwIST (black) over algorithm iterations; and scenario: (**a**) DCT blocked, (**b**) DCT whole, (**c**) DWT blocked, and (**d**) DWT whole.

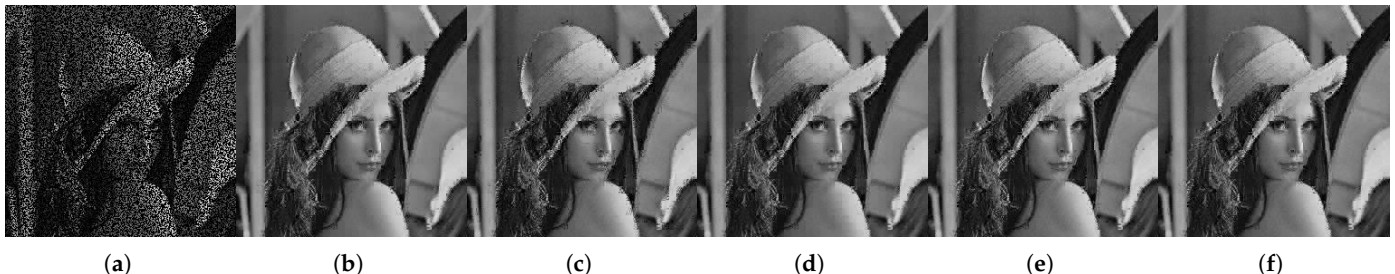

(**a**) (**b**) (**c**) (**d**) (**e**) (**f**)

**Figure 6.** Sparse reconstruction results for blocked DCT scenario: (**a**) CSed image ($M = 0.4$); (**b**) reconstructed by the TwIST (MSE = 191.46); (**c**) reconstructed by the SALSA (MSE = 273.30); (**d**) reconstructed by the NESTA (MSE = 181.81); (**e**) reconstructed by the YALL1 (MSE = 189.69); and (**f**) reconstructed by the FICI-TwIST (MSE = 176.99).

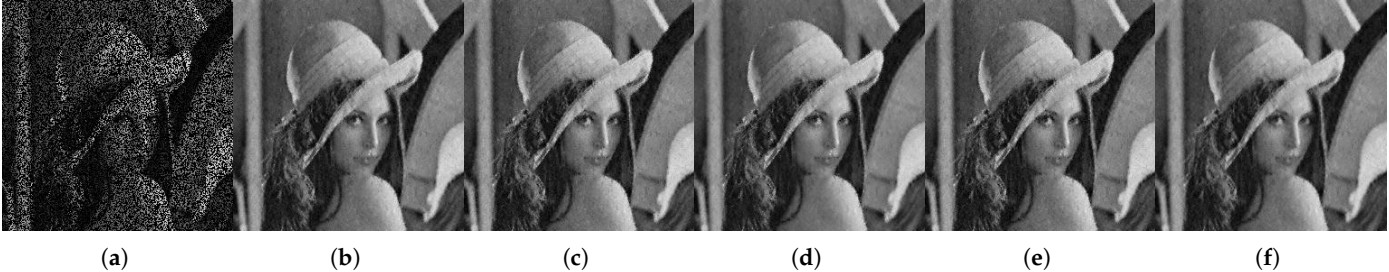

(**a**) (**b**) (**c**) (**d**) (**e**) (**f**)

**Figure 7.** Sparse reconstruction results for whole DCT scenario: (**a**) CSed image ($M = 0.4$); (**b**) reconstructed by the TwIST (MSE = 145.45); (**c**) reconstructed by the SALSA (MSE = 150.94); (**d**) reconstructed by the NESTA (MSE = 135.88); (**e**) reconstructed by the YALL1 (MSE = 151.24); and (**f**) reconstructed by the FICI-TwIST (MSE = 143.91).

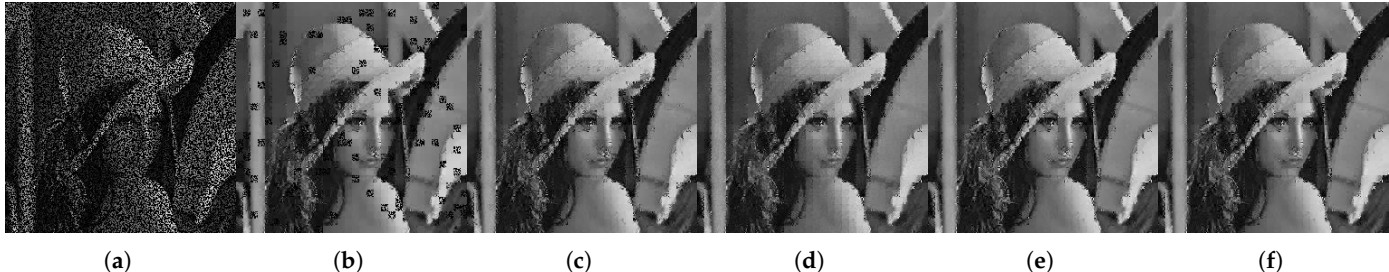

(**a**)  (**b**)  (**c**)  (**d**)  (**e**)  (**f**)

**Figure 8.** Sparse reconstruction results for blocked DWT scenario: (**a**) CSed image ($M = 0.4$); (**b**) reconstructed by the TwIST (MSE = 1093.89); (**c**) reconstructed by the SALSA (MSE = 527.58); (**d**) reconstructed by the NESTA (MSE = 447.90); (**e**) reconstructed by the YALL1 (MSE = 481.04); and (**f**) reconstructed by the FICI-TwIST (MSE = 432.41).

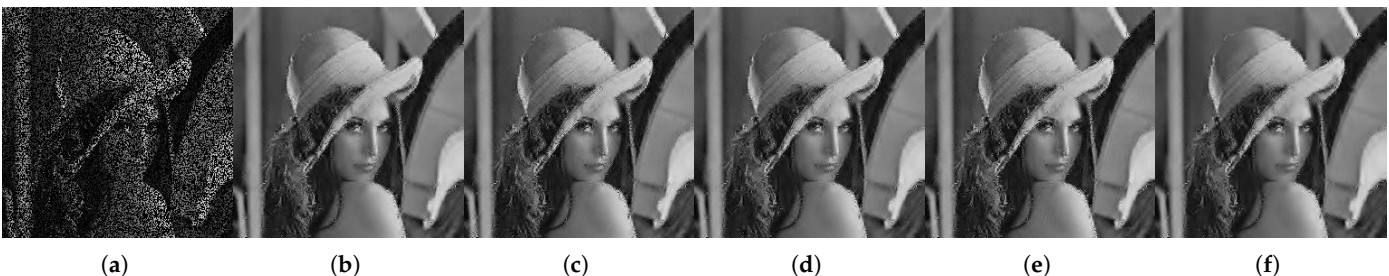

(**a**)  (**b**)  (**c**)  (**d**)  (**e**)  (**f**)

**Figure 9.** Sparse reconstruction results for whole DWT scenario: (**a**) CSed image ($M = 0.4$); (**b**) reconstructed by the TwIST (MSE = 190.04); (**c**) reconstructed by the SALSA (MSE = 210.38); (**d**) reconstructed by the NESTA (MSE = 182.59); (**e**) reconstructed by the YALL1 (MSE = 198.14); and (**f**) reconstructed by the FICI-TwIST (MSE = 186.17).

We have also performed simulations on two more standard test images: Barbara and Cameraman with a same simulation setup. The obtained results are not significantly different to the previously discussed results for Lenna, as can be seen in Figures 10–13.

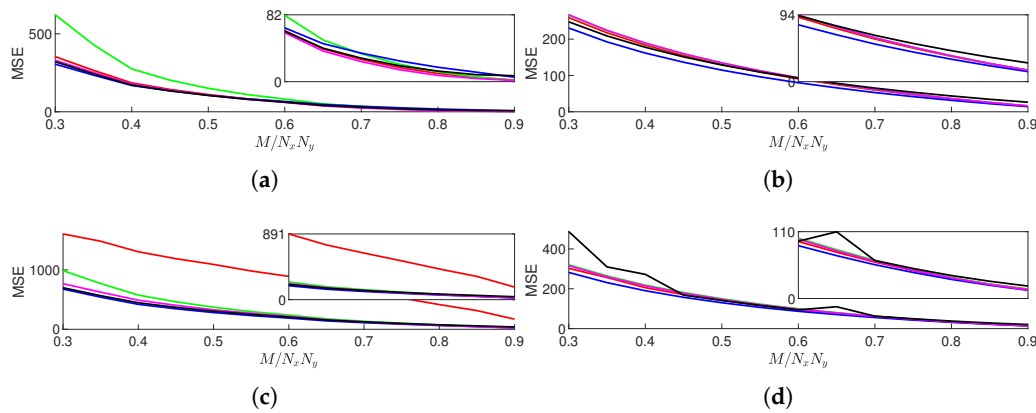

**Figure 10.** Average reconstructed MSE values for Barbara and TwIST (red), SALSA (green), NESTA (blue), YALL1 (magenta), and FICI-TwIST (black) over $M/N_xN_y$ range; and scenario: (**a**) DCT blocked, (**b**) DCT whole, (**c**) DWT blocked, and (**d**) DWT whole. Range of the zoom inset is $M/N_xN_y \in [0.6, 0.9]$.

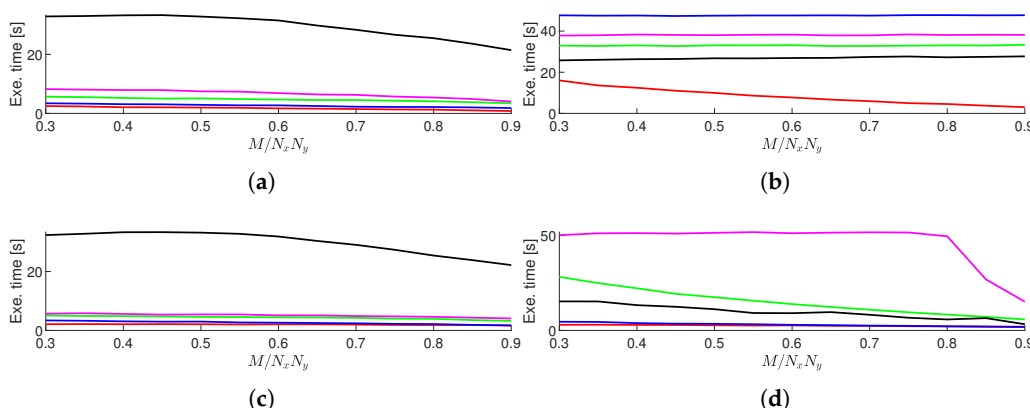

**Figure 11.** Average algorithm execution time for Barbara and TwIST (red), SALSA (green), NESTA (blue), YALL1 (magenta), and FICI-TwIST (black) over $M/N_xN_y$ range; and scenario: (**a**) DCT blocked, (**b**) DCT whole, (**c**) DWT blocked, and (**d**) DWT whole.

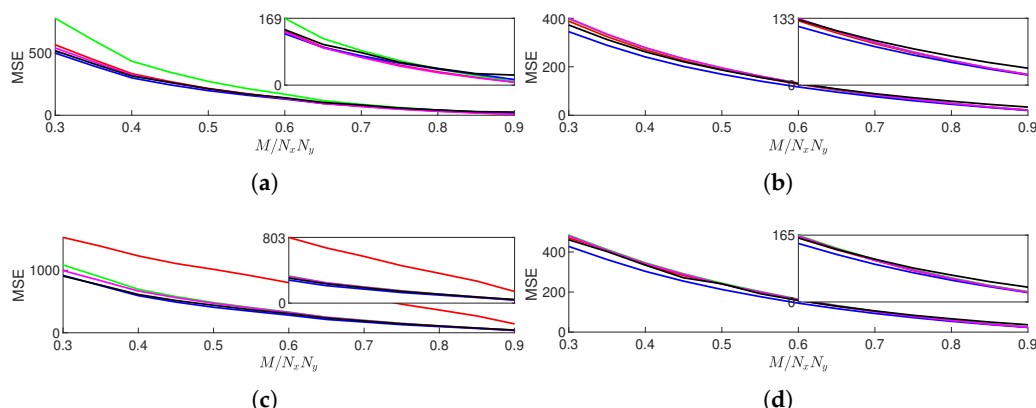

**Figure 12.** Average reconstructed MSE values for Cameraman and TwIST (red), SALSA (green), NESTA (blue), YALL1 (magenta), and FICI-TwIST (black) over $M/N_xN_y$ range; and scenario: (**a**) DCT blocked, (**b**) DCT whole, (**c**) DWT blocked, and (**d**) DWT whole. Range of the zoom inset is $M/N_xN_y \in [0.6, 0.9]$.

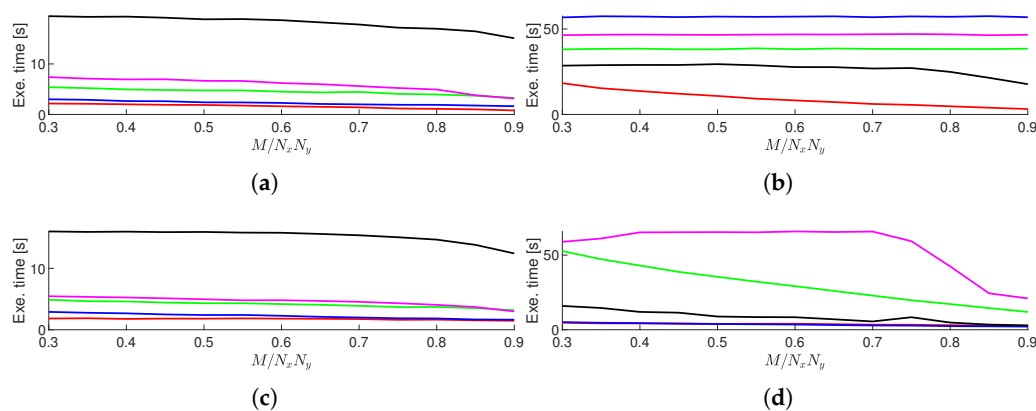

**Figure 13.** Average algorithm execution time for Cameraman and TwIST (red), SALSA (green), NESTA (blue), YALL1 (magenta), and FICI-TwIST (black) over $M/N_xN_y$ range; and scenario: (**a**) DCT blocked, (**b**) DCT whole, (**c**) DWT blocked, and (**d**) DWT whole.

## 4. Conclusions

In this paper, we have demonstrated the effectiveness of the sparsity-based image recovery. In our test scenarios, images with up to 60% of missing pixels have been successfully recovered (which corresponds to the CS ratio of $M/N_x N_y = 0.4$). More importantly, we have derived the matrix form of the wavelet transform, which is based on the CDF97 wavelet, achieving a significant reduction (by a factor of 10) in the considered sparse reconstruction algorithm's execution times as compared to the function-based transformation. This improvement is achieved by precalculating the transformation coefficients, keeping them in the memory, and using them multiple times through the reconstruction process. The function-based transformation, on the other hand, recalculates the coefficients each time it is called. Due to this fact, the matrix-based transformation requires additional memory space, which inherently limits its applicability when dealing with larger block sizes.

Such a matrix has allowed us to extend our previous research findings, in which we had proposed a sparse reconstruction algorithm, denoted as the FICI-TwIST, with its main advantage being its adaptivity, achieved by taking variable iterative thresholding steps. Our simulations have shown that even in the scenario in which all individual parameters of the considered state-of-the-art algorithms were fine-tuned, the FICI-TwIST algorithm runs very competitively, even outperforming the others in specific cases. The FICI-TwiST method's downside lies in the fact that it requires constant recalculation of the mean value and the standard deviation, which is relatively time-consuming. This is why the FICI-TwIST algorithm performs better with larger block sizes, a fact that is, interestingly, opposite to the applicability of the previously discussed matrix-based transformation.

In order to find the optimal block size, more detailed simulations with variable block sizes would have to be performed, taking into account both the reconstruction accuracy and the execution time. Such simulations could also be useful for additional comparison of the DCT vs. the DWT domains' performances. In addition, the relative $\ell_2$-norm change has proven to be an inadequate algorithm-exit criterion; thus, designing a more image-specific criterion would further decrease the reconstruction execution times of all the considered algorithms. All of the above outlined issues are the focus of our ongoing and future research and will be thoroughly addressed in our future publications.

**Author Contributions:** Conceptualization, I.V. and V.S.; methodology, I.V.; software, I.V.; validation, I.V. and V.S.; formal analysis, I.V.; investigation, I.V.; resources, I.V.; data curation, I.V.; writing—original draft preparation, I.V.; writing—review and editing, V.S.; visualization, I.V.; supervision, V.S.; project administration, V.S.; funding acquisition, V.S. All authors have read and agreed to the published version of the manuscript.

**Funding:** This work has been fully supported by the University of Rijeka under project number UNIRI-TEHNIC-18-67.

**Institutional Review Board Statement:** Not applicable.

**Informed Consent Statement:** Not applicable.

**Data Availability Statement:** The data presented in this paper are available on request from the corresponding author.

**Conflicts of Interest:** The authors declare no conflict of interest.

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
