# Peer review of "A Fast Intersection of Confidence Intervals Method-Based Adaptive Thresholding for Sparse Image Reconstruction Using the Matrix Form of the Wavelet Transform"

_information, doi:10.3390/info15020071_

Round 1

Reviewer 1 Report

Comments and Suggestions for Authors

In the paper, the authors extend their previous work on the FICI-TwIST algorithm. In the extension, the authors derive a single-wrapped Cohen-Daubechies-Feauveau 9/7 matrix, which allows for testing the algorithm in both the DCT and DWT domains. The use of this matrix decreases the execution time of the algorithms. The paper presents interesting results, and in general, it is well written. There are only some minor issues that I want to raise:
1. In paper titles, one should avoid using abbreviations. Thus, the ICI abbreviation in the title should be expanded to its full form.
2. We cannot start a subsection right after starting a section. There should be at least one paragraph of text. Thus, the authors should add such a paragraph between 2 and 2.1.
3. Line 78, the c function is not working on R^2 but on R^{N_x N_y}.
4. In (9), the authors do not use bold font for phi_{k, n}^{(N)} in the matrix, but in the definition which we find in (8), they use a bold font. The notation should be consistent.
5. In (11), the authors use two times "otherwise". This is a mistake because the first use of "otherwise" creates all cases that are not satisfied earlier, so there cannot be any "otherwise" to "otherwise".
6. Line 141, the authors write m_{1, 2} = \pm n - 2k. Which value n - 2k, -(n - 2k) corresponds to which variable m_1, m_2?
7. In many places in the text, the authors use "indexes". The correct plural form of "index" is "indices".
8. Algorithm 1 contains a pseudocode of the FICI-TwIST. Each algorithm should have Input and Output so that the reader knows what should be given to the algorithm and what will be returned by the algorithm. In the current version of the algorithm, the Output is missing. Moreover, the R variable is not initialized in any step of the algorithm, so the while loop cannot check the condition R < R_c because R is undefined.
9. Line 210 and in other places in the text, "lenna" -> "Lena".
10. Line 287, "barbara" -> "Barbara".
11. The authors show only the MSE plots for the other test images. They should also include the time plots.
12. In Conclusion, the authors claim that the proposed method successfully recovered an image with up to 60% of missing pixels. In Sec. 3, the authors do not give any information on the percentage of missing pixels. Moreover, one could ask which threshold value of the percentage of missing pixels causes that the method cannot recover the image?

Author Response

We would like to thank the reviewer for their constructive comments. We have made the requested changes, as detailed below.

  1. We have replaced the ICI abbreviation with the full name in the paper title, and slightly modified the title to accommodate for this change.
  2. We have added a short paragraph between Sections 2 and 2.1, explaining the organization of the Section in question.
  3. We have changed the inline expression accordingly.
  4. We have un-bolded phi_{k, n}^{(N)} in (8).
  5. We have replaced two “otherwise”-s in (11) with the opposite conditions, in order to avoid confusion.
  6. We have separated m_1 and m_2 to avoid confusion.
  7. We have replaced all instances of “indexes” with “indices”.
  8. We have added “Output: x_v” to the pseudo-code, and initialized the value of R to -1 (the value which satisfies the condition of the while loop regardless of the input R_c value).
  9. We have replaced all instances of “lenna” with “Lenna”.
  10. We have replaced all instances of “barbara” with “Barbara”.
  11. We have added the requested plots, and slightly modified the introducing text.
  12. The claim of 60% comes from the CS ratio of 0.4. If M/N_xN_y is 0.4, it follows that there is only 40% of the original pixels, thus 60% is missing. We have added a short note in the conclusion on this point. To answer the reviewer’s second question, throughout the paper we have provided several plots which show the reconstruction accuracy (MSE) as a function of CS ratio, which we have varied from 0.1 to 0.9 with a step of 0.05. The original claim of 60% comes from the authors subjective assessment of the resulting images. The CS ratio of 0.35 is still visually acceptable, but the image quality rapidly decreases for lower ratios. On the other hand, with larger CS ratios, the images become visually undistinguishable, as stated in the original manuscript. For the claimed CS ratio, the reconstructed images visually look fine, but one can start to observe some of the reconstruction defects. Of course, this depends on a definition of “looks fine”, so one can argue for a slightly higher or lower threshold ratio.

Reviewer 2 Report

Comments and Suggestions for Authors

The authors enhance their prior work in two-step iterative shrinkage thresholding (TwIST) algorithm via a fast intersection of confidence interval method, demonstrating improvements in the accuracy of signal reconstructions with relatively minor increases in run time as a result. The developed method does not require use of a tradeoff parameter unlike many other techniques, and a single wavelet transformation matrix is further proposed in order to further improve computational tractability. An ample range of CS ratios are explored, utilizing diverse test images and both DCT/DWT.

The paper is well written, sufficiently detailed, and effectively organized. Results appear scientifically sound and amply supported. The figures included are clear and wisely chosen. The following are relatively small issues that would be useful to address in revision:

-A more detailed review of the limitations of the developed methodology in the conclusions would be useful to provide a balanced perspective.  

-  It is appreciated how the authors transparently and fairly point out “All algorithm parameters have been fine-tuned for best performance with the CS ratio of 0.4, which in the hindsight, did not highlighted the main advantage of the proposed algorithm: its adaptivity; however, ’sabotaging’ other algorithms would be, lightly said, dishonest.”

Does the comparison change much if say a representative higher (or lower) CS ratiois used?

Comments on the Quality of English Language

-There are a few minor grammatical issues/typos, but overall the writing is clear. A few small errors are pointed out below:

Pg 2:   denoted as the FICI-TwIST, in which we have remove the dependency on the user defined threshold value which controls a trade-off between the solution accuracy and the convergence rate of a considered class of the reconstruction algorithms.

-> remove should be changed to removed

pg 8: When image is divided into blocks, SALSA (for the DCT) and TwIST (for the DWT) run significantly worst than the other algorithms for lower CS ratios, however, in general all algorithms have very similar reconstruction performances, likely due to the mentioned parameter fine-tuning.

->When image should be changed to When the image.

Author Response

We would like to thank the reviewer for their constructive comments. We have made the requested changes, as detailed below.

- We have expanded the conclusion with a discussion about the limitations of both the matrix-based transformation and the FICI-TwIST algorithm.

- The parameters have been optimised for CS ratio of 0.4 as this was shown to be a threshold value that leads to a successful image reconstruction. To answer the reviewer’s question, we have optimised the parameters and performed an additional performance test for just one scenario (DWT whole) for CS ratio of 0.8, and use it over the entire CS range (without averaging over 50 runs). The obtained results are relatively similar to those reported in this paper, however, without averaging over multiple runs it is hard to give definite answer. Also, different optimisation algorithms that we have been testing in the past (e.g. PSO) have not produced a set of parameters that would result in lower MSE values than those obtain using the parameters chosen manually (as done in this paper), preventing us to optimize algorithm parameters for each CS ratio separately.

- We have corrected the grammatical issues/typos.